# Structural and Functional Responses of the Heteromorphic Leaves of Different Tree Heights on *Populus euphratica* Oliv. to Different Soil Moisture Conditions

**DOI:** 10.3390/plants11182376

**Published:** 2022-09-12

**Authors:** Juntuan Zhai, Zhijun Li, Jianhua Si, Shanhe Zhang, Xiaoli Han, Xiangxiang Chen

**Affiliations:** 1College of Life Sciences, Tarim University and Key Laboratory of Protection and Utilization of Biological Resources in Tarim Basin, Xinjiang Production & Construction Corps and Research Center of Populus Euphratica, Alar 843300, China; 2Northwest Institute of Eco-Environment and Resources, Chinese Academy of Sciences, Lanzhou 730000, China

**Keywords:** *Populus euphratica* Oliv., heteromorphic leaf, morphological structure, physiological, soil drought

## Abstract

*Populus euphratica* Oliv., a pioneer species of desert riparian forest, is characterized heterophylly. To understand the adaptation strategies of the heteromorphic leaves of *P. euphratica* to soil drought, we assessed the structural and functional characteristics of the heteromorphic leaves at different heights in suitable soil moisture conditions (groundwater depth 1.5 m) and drought conditions (groundwater depth 5 m), which include morphology, anatomical structure, photosynthetic capacity, water use efficiency, osmotic adjustment capacity, and endogenous hormones. These results indicate that leaf area, leaf thickness, fence tissue, palisade-to-sea ratio, main vein xylem area, vessel area, net photosynthetic rate, transpiration rate, and proline, MDA, IAA, GA_3_, and ZR contents showed a positive correlation with the tree height under the two soil moisture conditions, but leaf shape index, leaf water potential (LWP), and ABA content showed a decreasing trend. In addition, the main vein vascular bundle area, main vein xylem area, and contents of malondialdehyde, ABA, GA_3_, and IAA were significantly greater under soil drought conditions than normal soil water content. Under soil drought stress, the heteromorphic leaves of *P. euphratica* showed more investment in anatomical structure and greater water use efficiency, proline, and hormone contents, and synergistic changes to maintain high photosynthetic efficiency. This is an adaptation strategy to water stress caused by soil drought and tree height changes.

## 1. Introduction

The structural and functional traits of plant leaves can directly reflect the adaptability of plants to the growth environment and show different adaptation strategies in diverse environments [1,2]. For example, plants often reduce the leaf area to adapt to arid climates [3]. The leaf area of *Parashorea chinensis* Wang Hsie. decreases to cope with the water stress as the tree height increases [4]; however, *Fagus sylvatica* forests showed an increase in leaf size and a reduction in the number of leaves under drought [5]. Some plants increase their leaf area to grow faster and make greater use of ground resources [6,7]. Leaves in the lower canopy have relatively higher specific leaf areas [8,9]. The key mechanism underlying tree death is the destruction of the hydraulic supply system due to severe water stress [10,11,12]. The development of the vascular tissue, xylem, phloem, and palisade tissue thickness significantly affects drought resistance due to water transport functions and photosynthesis sites [13,14]. For example, the palisade tissue thickness, ratio of the palisade to spongy tissue, and vascular bundle density of *Parashorea chinensis* increased with tree height and the leaf showed a more robust xerophytic structure [4]. Leaf thickness, palisade tissue, and other xerophytic structures of the heteromorphic leaves of *P. euphratica* changed with tree height [15]. However, the changes in heteromorphic leaf structure traits with tree height and the adaptation strategies under soil drought conditions in *Populus euphratica* Oliv. are still unclear.

Drought stress also limits photosynthesis [16,17]. Knowledge of the photosynthesis of plant leaves in different habitats provides information on the ability of plants to adapt to different environmental conditions [18,19,20,21,22]. Generally, drought stress significantly reduces the photosynthetic rate of plants [23,24]; however, some species reduce leaf water potential during drought stress by keeping the stomata open for continuous photosynthesis despite the high risk of hydraulic failure [10,12]. Meanwhile, the effects of low-intensity drought stress can be compensated by increasing nutrient availability and changing source-sink balance [25], It is widely acknowledged that new assimilates are more allocated to the root system under drought [26,27]. Higher plants accumulate osmolytes such as proline and malondialdehyde after being exposed to water stress [28]. These osmolytes affect the photosynthetic capacity of plants under drought stress [15,17,29]. Endogenous hormones enhance the chances of plant survival in adverse environments by inducing the accumulation of soluble osmotic substances [30,31]. For plants in arid and semi-arid regions, water use efficiency is another important factor limiting photosynthesis [24,32]. Increased water use efficiency is beneficial to photosynthesis of the upper leaves of the canopy [33], and leaf stable carbon isotope ratio (δ^13^C) is a proxy for water-use efficiency [34]. Under suitable soil moisture conditions, the photosynthetic capacity of *P. euphratica* heteromorphic leaves increases with an increase in tree height and synergistically enhances δ^13^C value and osmotic adjustment ability [15]. However, under soil drought conditions, the functional characteristics such as changes in photosynthetic capacity, water use efficiency, osmotic adjustment capacity, and adaptation strategies of *P. euphratica* heteromorphic leaves with increasing tree height are still unclear.

Many riparian trees are heavily dependent on groundwater [24,35,36,37,38]. The depth of groundwater affects its growth, composition, and succession [39,40,41]. Natural *P. euphratica* forests, similar to many riparian tree species, depend on groundwater for survival and growth and have been used as a model species for research on drought [17,42,43]. *P. euphratica* grows at a depth range of 0.5–4.71 m at groundwater level and in the range of 4.71–8.62 m under stress. When the local water level is greater than 8.62 m, it shows a size decline [44]. Xu’s research shows that the groundwater level of *Populus euphratica* under water stress appears at 5.0 m [45]. *P. euphratica* has heteromorphic leaves that are related to the developmental stage and environmental conditions. It is considered to be an adaptive feature formed during the long-term evolution of *P. euphratica* [46]. Current research on the heteromorphic leaf properties of *P. euphratica* has mainly focused on comparing the morphological structure, photosynthetic water physiology, and osmotic adjustment ability of different heteromorphic leaves on the same plant [47,48,49,50,51] and between different tree ages [15,51,52,53,54,55]. However, there are no reports on the response of the heteromorphic leaves of *P. euphratica* to changes in the morphological structure and physiological functions under soil drought stress conditions. We hypothesized that: (1) *P. euphratica* shows differences in the morphological and structural characteristics, photosynthetic capacity, water use efficiency, endogenous hormones, and osmotic adjustment ability of heteromorphic leaves under suitable soil water and drought stress conditions. (2) *P. euphratica* shows different ecological adaptation strategies in response to different soil moisture conditions. We analyzed how *P. euphratica* responds to water stress caused by changes in tree height and under soil drought stress to understand the relationship between heteromorphic leaves and adaptive evolution. Considering increasing drought under climate change, knowledge of the effects of different soil moisture conditions on the growth of *P. euphratica* is crucial for ecological restoration and afforestation efforts in arid areas.

## 2. Results

### 2.1. Change in Morphological Traits of Heteromorphic Leaves with Tree Height

Under the conditions of suitable soil moisture and drought stress, the heteromorphic leaf LA showed an increasing trend with the increase in tree height (Figure 1b), the LI at 2 m was significantly larger than other heights, and the LT at the crown height (6 m, 8 m, and 10 m) was significantly larger than that at the lower part (Figure 1a,c). Under soil drought conditions, the LA at tree heights of 2, 4, and 6 m was significantly greater than that under suitable water conditions, and the SLA of 4 m tree heights was significantly lower than that under suitable moisture conditions (Figure 1b,d).

### 2.2. Change in Structure Traits of the Heteromorphic Leaves with Tree Height

Under the two soil moisture conditions, the heteromorphic leaves showed an increasing PT, PSR, MVBA, MXA, XA/VBA, and VA with an increase in tree height. There were significant (*p* < 0.05) differences in the structural index parameters between the tree heights at 2 m and 10 m. At the same tree height, MVBA, MXA, and VA under soil drought stress were significantly (*p* < 0.05) larger than those under suitable soil moisture conditions. The PSR at 6, 8, and 10 m under soil drought conditions was significantly (*p* < 0.05) greater than that under suitable moisture conditions (Table 1). The results showed that under the two soil moisture conditions, the anatomical structures of the heteromorphic leaves gradually changed as the tree height increased. Under soil drought conditions the leaf structure traits were more developed than under suitable moisture conditions.

### 2.3. Change in Photosynthetic Physiology of Heteromorphic Leaves with Tree Height

Under the two soil moisture conditions, the Pn, Tr, and Gs of the heteromorphic leaves increased significantly (*p* < 0.05) as the tree height increased (Figure 2a–d); At the same tree height, the Pn and Tr at 6, 8, and 10 m under soil drought stress conditions were significantly greater than those under suitable soil moisture conditions (*p* < 0.05). The results showed that under the two soil moisture conditions, the photosynthetic capacity of the heteromorphic leaves increased with tree height.

### 2.4. Change in Water Use Efficiency of Heteromorphic Leaves with Tree Height

As the tree height increased, under the two soil moisture conditions, the δ^13^C values of the heteromorphic leaves showed an increasing trend, the LWP of the heteromorphic leaves showed a decreasing trend (Figure 3b,d), but the RWC and instantaneous water use efficiency did not significantly change (Figure 3a,c). At the same tree height, the RWC, LWP, and WUE_i_ under soil drought stress were significantly lower than that under suitable soil water conditions, but the δ^13^C values of the heteromorphic leaves at each tree height under soil drought stress were significantly greater than those under suitable soil water conditions (*p* < 0.05). Although the water conditions of heteromorphic leaves under soil drought conditions were not as satisfactory as those under suitable water conditions, the long-term water use efficiency was significantly higher.

### 2.5. Change in Physiological Characteristics of Heteromorphic Leaves with Tree Height

Under the two soil moisture conditions, the Pro, MDA, and SP in the heteromorphic leaves increased as the tree height increased (Figure 4a–d). At the same tree height, under soil drought stress conditions, the Pro content at 2, 8, and 10 m tree heights, the SP content at 2, 6, 8, and 10 m tree heights, and the MDA content at each sampling height were significantly greater than those under suitable soil moisture conditions (*p* < 0.05) (Figure 4a,b,d). At a height of 10 m, the osmotic adjustment ability under drought conditions is stronger than that under suitable water conditions.

### 2.6. Change in Endogenous Hormones of Heteromorphic Leaves with Tree Height

The content of GA_3_, IAA, and ZR in heteromorphic leaves showed an increasing trend with an increase in tree height under the two soil moisture conditions. In contrast, the content of ABA showed a decreasing trend with an increase in tree height (Figure 5a–d). The contents of ABA, GA_3_, and IAA at each tree height under drought stress conditions were significantly greater than those under suitable soil moisture conditions (*p* < 0.05) (Figure 5a–c). The results showed that the endogenous hormone content of the heteromorphic leaves under soil drought conditions was significantly higher than that under suitable water conditions.

### 2.7. Correlation Analysis and Regression Analysis of Heteromorphic Leaves Structure and Function Traits and Tree Height

Correlation analyses (Appendix A) showed that under the two soil moisture conditions, the LA, LT, PT, PSR, MVBA, MXA, XA/VBA, VA, Pn, Tr, Gs, δ^13^C, Pro, MDA, SP, and GA_3_, IAA, and ZR contents were significantly positively correlated with tree height whereas LI and ABA were significantly negatively correlated with tree height. PT, MVBA, MXA, XA/VBA, VA, Pn, Tr, δ^13^C, Pro, and GA_3_, IAA, and ZR contents with LA were significantly positively correlated. Under different soil moisture conditions, the structural and functional traits of the heteromorphic leaves changed regularly with an increase in tree height.

According to Table 2, the fit of heteromorphic LI, LA, LT, PT, MVBA, MXA, and VA was adequate (R^2^ > 0.4). The absolute value of the regression coefficients of LI, LA, and LT under soil drought stress was less than that under suitable soil moisture conditions. However, the absolute values of the regression coefficients of PT, MVBA, MXA, and VA under soil drought stress were greater than those under suitable soil moisture conditions. This information shows that under suitable soil moisture conditions, an increase in tree height had a greater impact on LI, LA, and LT. Under soil drought stress, an increase in tree height had a greater impact on the PT, MVBA, MXA, and VA. It showed that under different soil moisture conditions, the morphological structure of heteromorphic leaves differed as the tree height increased.

According to Table 3, the Pn, Ci, δ^13^C, GA_3_, IAA, and ZR were adequately fitted (R^2^ > 0.4). The absolute value of the regression coefficient under soil drought stress was greater than that under suitable soil moisture conditions. An increase in tree height had a relatively greater impact on the net photosynthetic capacity, proline, and hormone content of the heteromorphic leaves under soil drought stress.

## 3. Discussion

### 3.1. Responses of Heteromorphic Leaves Structure Traits to Soil Drought Stress

Plant canopy height is an essential attribute of the forest ecosystem [56,57]. It has an important influence on the biomass allocation of branches and leaves [58]. Different canopies of trees often produce vertical gradients due to the projection of sunlight, which causes differences in the anatomical structure and metabolism of the leaves on the upper and lower layers of the canopy; for example, the specific leaf weight of the *Acer saccharum* increased with an increase in the crown [59,60,61]. The heteromorphic leaves of *P. euphratica* differ at different developmental stages and canopy heights under suitable soil moisture conditions [15]. Our findings showed that under suitable soil moisture conditions and drought stress conditions, the LA, LT, PT, PSR, MVBA, MXA, XA/VBA, and VA of the heteromorphic leaves *P. euphratica* increased as the height of the canopy increased. However, the MVBA, MXA, and VA at each tree height under soil drought conditions were significantly larger than those under suitable soil moisture conditions. In addition, the PSR and XA/VBA at 6, 8, and 10 m tree heights under soil drought conditions were significantly greater than those under suitable moisture conditions. This information suggests that *P. euphratica* responds to soil drought and water stress at different tree heights by changing the thickness of the heteromorphic leaves PT and the MVBA to enhance drought resistance.

A previous study reported that taller trees have longer vertical water-conducting paths from the roots to the tree crowns, and require greater tension and water gradients [62]. However, increased leaf water stress due to gravity and path length resistance may eventually limit leaf expansion and photosynthesis to promote further growth in height [63]. A study with *Eucalyptus amygdalina* found that leaf size had an important influence on the canopy–water relationship [64]. The reduced area of transpiration and photosynthesis supported the plant in coping with water stress as the tree aged and its height increased [64]. The leaf area of the heteromorphic leaves of *P. euphratica* increased as tree height increased; the leaf area was significantly positively correlated with the ratio of palisade tissue thickness to spongy tissue and main vein xylem/main vascular bundle area. These results showed that increased leaf area at canopy height impacts the enhancement of leaf xerophytic structures. *P. euphratica* adapts by increasing leaf area with tree height to improve photosynthetic and water transport efficiencies.

### 3.2. Responses of Heteromorphic Leaves Functional Traits to Soil Drought Stress

Leaf distribution and leaf morphology affect canopy photosynthesis and total primary productivity [65]. Our previous research showed that photosynthetic capacity and water use efficiency of the heteromorphic leaves of *P. euphratica* under suitable soil moisture conditions were different at different development stages and canopy heights at the same development stage [15]. This study showed that under suitable soil water conditions and drought stress conditions the net photosynthetic rate and transpiration rate of *P. euphratica* heteromorphic leaves increased significantly with an increase in tree height. The photosynthetic capacity and water use efficiency were different at the canopy height in this study. The heteromorphic leaves’ net photosynthetic and transpiration rates were significantly positively correlated with the height of the canopy, leaf area, and palisade tissue thickness, under soil drought conditions, the net photosynthetic and transpiration rates at 6, 8, and 10 m tree heights were significantly greater than those under suitable soil moisture conditions. The long-term water use efficiency was significantly greater under soil drought conditions than under normal moisture conditions, but the leaf water potential and relative water content were the opposite. High evaporating demand and low soil water content induce a decrease in water potential all along the pathway [66]. Leaf water potential decreases under drought conditions relative to a well-watered environment, thus reaching a lower leaf water potential and relative water content with rising evaporative demand and maintaining the driving force for water flow to leaves [67]. Increased photosynthesis as a consequence of a variety of environmental stressors has been reported in some studies [68,69] and is considered a tool used by trees to enact acclimation processes. With higher photosynthetic rates under drought conditions, we think maybe because mild drought activates the stress response and promotes the long-distance transport of sucrose, the increase in sucrose transport speed is conducive to alleviating the photoinhibition caused by the accumulation of photosynthetic products, to improve photosynthetic efficiency [26,27]. In addition, the increase in leaf area and palisade tissue thickness also provides expenses for the improvement in photosynthetic efficiency. The organic coupling of photosynthetic physiology, water physiology, and morphological structure characteristics contributes to this result.

In higher plants, water stress increases the accumulation of osmolytes such as proline and malondialdehyde to cope with the pressure caused by water shortage [28]. The osmotic adjustment ability of *P. euphratica* broad-ovate leaves is greater than that of other leaves [50]. This ability increased as the tree height increased in the heteromorphic leaves of *P. euphratica* under suitable soil moisture conditions [15]. In our study, under the two soil moisture conditions, the osmotic adjustment ability of heteromorphic leaves increased with tree height. The content of proline, malondialdehyde, and soluble proteins increased as the tree height increased. A significant positive correlation was found among leaf area, net photosynthetic rate, and transpiration rate. These results showed that *P. euphratica* increases the investment in the leaf area at the top of the crown, and the content of osmotic adjustment substances also increases. Increasing the content of these substances can adjust the water absorption capacity of the leaves and the net photosynthetic and transpiration rates. The proline and malondialdehyde contents at 2, 8, and 10 m tree heights under soil drought stress conditions were significantly greater than those under suitable soil moisture conditions. This shows that the heteromorphic leaves of *P. euphratica* have greater osmotic adjustment ability under soil drought conditions.

Multiple biochemical traits are used as indicators of drought resistance. Abscisic acid (ABA) is one of the most critical hormones involved in the regulation of drought resistance [70,71,72,73]. When plants are under stress, an increased ABA content reduces leaf stomatal conductance, transpiration loss, and the absorption and fixation of CO_2_, which inhibits plant growth [74]. Under the two soil moisture conditions, the ABA content of *P. euphratica* heteromorphic leaves decreased as the tree height increased; concurrently, ABA content was significantly negatively correlated with leaf area and net photosynthetic rate. A synergy was found between the decrease in ABA content in the heteromorphic leaves of *P. euphratica,* the increase in leaf area, and the net photosynthetic rate in the crown leaves. Some studies have demonstrated that the increase in GA_3_, IAA, and ZR content can enhance the stress resistance of plants [75,76]. The decrease in GA_3_ and IAA content represents the transition of plant growth to a stress adaptation state [74]. Our research showed that the content of GA_3_, IAA, and ZR in *P. euphratica* heteromorphic leaves increased with tree height and enhances the stress resistance of the leaves. Similar findings have been reported previously [75,76]. In addition, hormones can induce the accumulation of soluble osmotic substances through the regulation of metabolism and enhance the survival of plants in adversity [31]. The content of GA_3_, IAA, and ZR in *P. euphratica* heteromorphic leaves was significantly positively correlated with the content of proline. GA_3_, IAA, and ZR could enhance the stress resistance of *P. euphratica* heteromorphic leaves by inducing the accumulation of proline.

Despite the two soil moisture conditions, the contents of the four endogenous hormones of *P. euphratica* heteromorphic leaves presented a similar trend with an increase in tree height. However, the contents of ABA, GA_3_, and IAA under drought conditions were significantly greater than those under suitable moisture conditions. This information suggests that increased GA_3_, IAA, and ZR content in the heteromorphic leaves under drought conditions could be regarded as an adaptation strategy to cope with water stress brought by soil drought and tree height through the synergistic changes in osmotic adjustment substances.

## 4. Materials and Methods

### 4.1. Study Area

The study area, located on the north-western margin of the Tarim Basin in Xinjiang Province of China, has a typical temperate desert climate. The average annual rainfall is approximately 50 mm, the potential evaporation reaches up to 1900 mm, the yearly average temperature is 10.8 °C, and the average annual sunshine duration is 2900 h. Two forests of *P. euphratica* were selected as sampling points for comparison. One is located in Shaya County with a groundwater level of 5 m and the other is on the banks of the Tarim River in Alar City with a groundwater level of 1.5 m. The meteorological measurements at the two sampling points were collected in July. The soil moisture content was measured on a volume of soil taken from the surface to a depth of 1 m (Appendix A).

### 4.2. Plant Samples and Sampling Method

The areas of the two sampling points are 92 hectares and 45 hectares, respectively. In each sampling point, 5 trees were selected with the characters of diameter at breast height of 15 cm, age of 12 years, and height of 10 m. Samples were obtained at heights of 2, 4, 6, 8, and 10 m starting from the base of the trunk of each tree. We collected 30 branches from each sampling layer, selected the fourth node leaf from each branch, and took 30 leaves in total. A portion of the leaves was quickly frozen with liquid nitrogen and stored at −80 °C to determine the content of proline, malondialdehyde, soluble sugar, and hormones; another sample of the leaves was fixed with formalin–alcohol–glacial acetic acid mixed fixative (FAA). The fixed samples were used to prepare tissue sections; a part of the leaves was used to determine the morphological indicators of abnormal leaves. Sampling was performed during the vigorous growth period of *P. euphratica* in mid-July.

### 4.3. Determination of the Morphological and Anatomical Structural Indexes of Heteromorphic Leaves

We used a scanner (MRS-9600TFU2, Shanghai, China) and the LA-S plant image analysis software to measure the leaf length, leaf width, and leaf area of *P. euphratica.* The leaf shape index (LI) was calculated based on the leaf length/width ratio. We used a vernier caliper to measure the thickness and take the average value to obtain the leaf thickness (LT). The collected leaves were processed for deactivation (10 min) at 105 °C and heated at 65 °C to a constant weight. After the sample reached a constant weight, the material was cooled to room temperature (25 °C) in a desiccator and weighed with an electronic balance with an accuracy of 0.001 g; the specific leaf area (SLA) was then calculated.

The leaf blade was cut transversely at its widest section. The material that retained the primary vein and leaf margin was selected and fixed in a formalin–acetic acid–alcohol (FAA) solution. Paraffin-embedded tissue sections (8 µm thick) were prepared, double-stained with safranin-fast green, and mounted in a neutral resin. Observation and determination of the palisade tissue (PT), main vein vascular bundle area (MVBA), main vein xylem area (MXA), and vessel area (VA) were performed using a Leica microscope (Leica DM4 B, Wetzlar, Germany). The ratio of palisade tissue to spongy tissue (PSR), main vein xylem, vascular bundle area ratio (XA/VBA), five fields of view per leaf, 20 values per field of view, and an average value of leaf anatomical structure parameters of five fields of view was taken as the parameter values of the anatomical structural index of each leaf.

### 4.4. Determination of Photosynthetic Water Physiological Indices of Heteromorphic Leaves

Branch shears were used to cut branches of the current year. The branches were wrapped immediately with fresh-keeping film to cover the cut ends. Photosynthetic gas exchange characteristics were measured with a portable photosynthesis system, LI-COR 6400XT (LI-COR, Lincoln, NE, USA) for the fourth fully expanded leaf between 09:00 a.m. and 12:00 a.m. on 25 and 28 July 2019. Light-saturated net photosynthesis rate (P_n_), stomatal conductance (g_s_), intercellular CO_2_ concentration (C_i_), and transpiration rate (E) were measured under the following conditions: leaf temperature 25 °C, relative air humidity 60%, ambient CO_2_ concentration 400 ± 5 μmol mol^−1^, and photosynthetic photon flux density 1250 μmol m^−2^ s^−1^.

At each canopy height, leaves at node 3 or 4 of 10 annual shoots were selected to determine the relative water content (RWC) and leaf water potential (LWP). We measured the fresh mass (FM), turgid mass (TM), and dry mass (DM) of each leaf. Then, RWC was calculated as follows: RWC = 100 (FM–DM)/(TM–DM). Leaf water potential (LWP) was measured at 12:00 a.m. using a portable plant water potential pressure chamber (600-EXP).

The photosynthetic physiological indices of the leaves were measured with the Li-6400 photosynthesis instrument, and the instantaneous water use efficiency (WUE_i_ = Pn/Tr) of the irregular leaves was calculated. The leaf samples were washed and air-dried, then dried at 65 °C for 36 h, pulverized with a grinder, and passed through a 90-mesh sieve to prepare test samples. The carbon isotope analysis of the plant samples was prepared using a glass vacuum system. The furnace temperature of the burner was controlled at 1000 °C. The system was evacuated and O_2_ was passed through. The porcelain spoon containing the sample of *P. euphratica* leaves was placed in the combustion tube in the high-temperature zone. After burning for 2 min, the CO_2_ gas was collected and purified by freezing. A stable gas isotope mass spectrometer (Thermo Fisher Scientific, Inc., Waltham, MA, USA) was used to analyze the carbon isotope composition (δ^13^C value) of the purified CO_2_ gas.

### 4.5. Determination of Physiological and Biochemical Indices of Heteromorphic Leaves

The acid ninhydrin method was used to determine the leaf proline content (µg/g). The MDA content (µmol/g) was determined with the thiobarbituric acid color method, and the SS content (mg/g) was determined with the anthrone colorimetry method. The Coomassie brilliant blue G-250 dyeing method was used to determine the SP content (mg/g) of leaves [15].

### 4.6. Determination of Hormone Content in Heteromorphic Leaves

We took the leaves (without petioles) at the fourth node of the branches, quickly froze them with liquid nitrogen, and stored them at −80 °C. The enzyme-linked immunoassay method was used to determine the content of abscisic acid (ABA) (ng/g·FW), gibberellin (GA_3_) (ng/g·FW), indoleacetic acid (IAA) (ng/g·FW), and zeatin riboside (ZR) (ng/g·FW). These measurements were performed at the China Agricultural University.

### 4.7. Statistical Analysis

We used a variance analysis to evaluate the differences in the structure and functional traits of the heteromorphic leaf-shaped leaves between different tree heights and different habitats using SPSS 18.0 software (Chicago, IL, USA). The significance level of the differences was set to *p* < 0.05. We used linear regression analysis to evaluate the impact of tree height changes on each trait. All data were normally distributed and single-peaked.

## 5. Conclusions

The increase in tree height under suitable soil moisture conditions has a large impact on the morphology of the heteromorphic leaves. Under soil drought conditions, increased tree height had a relatively significant effect on the anatomical structure, photosynthetic capacity, water use efficiency, osmotic adjustment capacity, and hormone content in the heteromorphic leaves, which showed greater water use efficiency, proline and hormone content, and synergistic changes to maintain high photosynthetic efficiency. Therefore, with regard to ecological restoration and afforestation efforts, the effects of different soil moisture conditions on the growth of *P. euphratica* should be considered in relation to climate change and consequent increasing droughts, to improve the rationality of irrigation.

## Figures and Tables

**Figure 1 plants-11-02376-f001:**
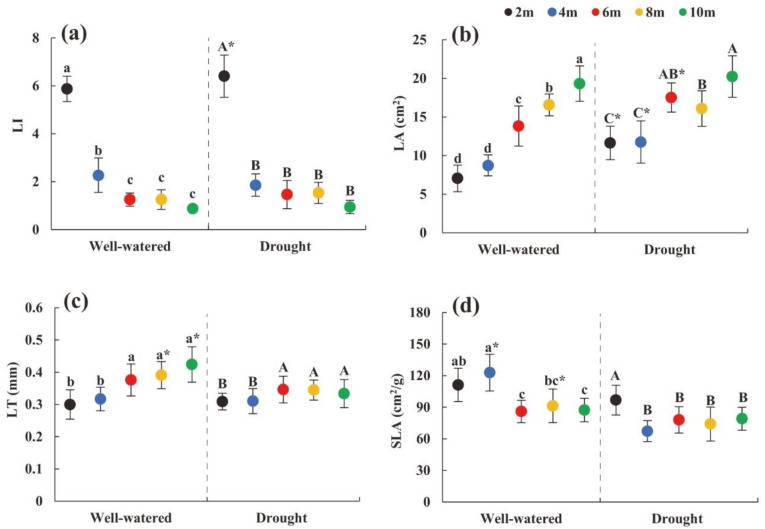
Changes in morphological characteristics of heteromorphic leaves with different tree heights under different soil moisture conditions. Note: (**a**–**d**) were changes in leaf index, leaf area, leaf thickness, specific leaf area with different tree heights under different soil moisture conditions. The black dot indicates the height of the 2 m tree, the blue dot indicates the height of the 4 m tree, the red dot indicates the height of the 6 m tree, the yellow dot indicates the height of the tree 8 m, and the green dot indicates the height of the 10 m tree; lowercase and uppercase letters represent the significance of the difference between different sampling heights under suitable soil moisture conditions and soil drought stress conditions, and * represents the comparison of the same height under different habitats (*p* < 0.05). LI: leaf index; LA: leaf area; LT: leaf thickness; SLA: specific leaf area.

**Figure 2 plants-11-02376-f002:**
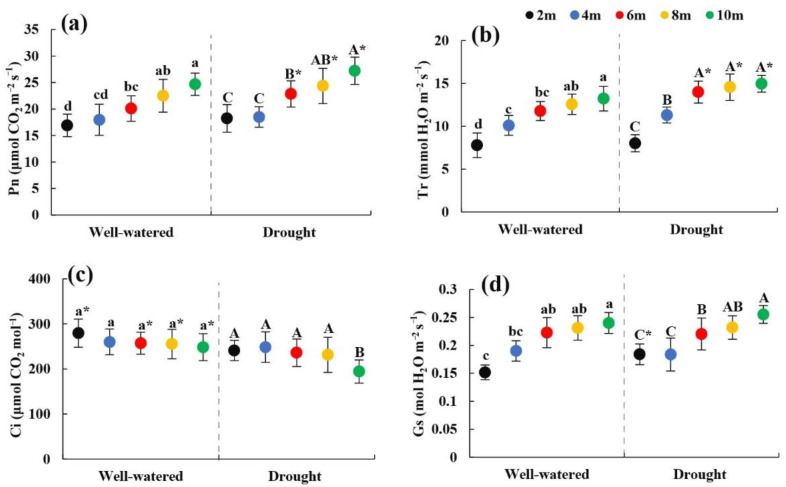
Changes in photosynthetic physiological parameters of heteromorphic leaves with tree height under different soil moisture conditions. Note: (**a**–**d**) were changes in photosynthetic rate, transpiration rate, intercellular CO_2_ concentration, stomatal conductance with different tree heights under different soil moisture conditions. The black dot indicates the height of the 2 m tree, the blue dot indicates the height of the 4 m tree, the red dot indicates the height of the 6 m tree, the yellow dot indicates the height of the tree 8 m, and the green dot indicates the height of the 10 m tree; lowercase and uppercase letters represent the significance of the difference between different sampling heights under suitable soil moisture conditions and soil drought stress conditions, and * represents the comparison of the same height under different habitats (*p* < 0.05). P_n_: photosynthetic rate; T_r_: transpiration rate; Gs: stomatal conductance; C_i_: Intercellular CO_2_ concentration.

**Figure 3 plants-11-02376-f003:**
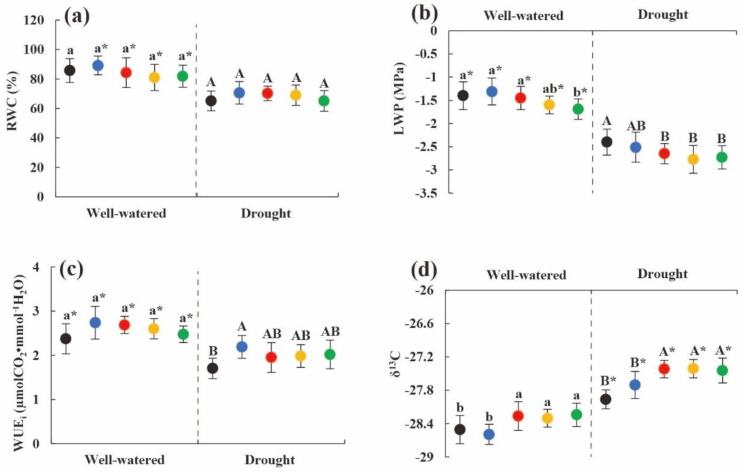
Changes in moisture content and utilization efficiency of heteromorphic leaves with tree height under different soil moisture conditions. Note: (**a**–**d**) were changes in relative water content, leaf water potential, instantaneous water use efficiency, stable carbon isotope value with different tree heights under different soil moisture condition. The black dot indicates the height of the 2 m tree, the blue dot indicates the height of the 4 m tree, the red dot indicates the height of the 6 m tree, the yellow dot indicates the height of the tree 8 m, and the green dot indicates the height of the 10 m tree; lowercase and uppercase letters represent the significance of the difference between different sampling heights under suitable soil moisture conditions and soil drought stress conditions, and * represents the comparison of the same height under different habitats (*p* < 0.05). RWC: relative water content; LWP: leaf water potential; WUE_i_: instantaneous water use efficiency; δ^13^C: stable carbon isotope value.

**Figure 4 plants-11-02376-f004:**
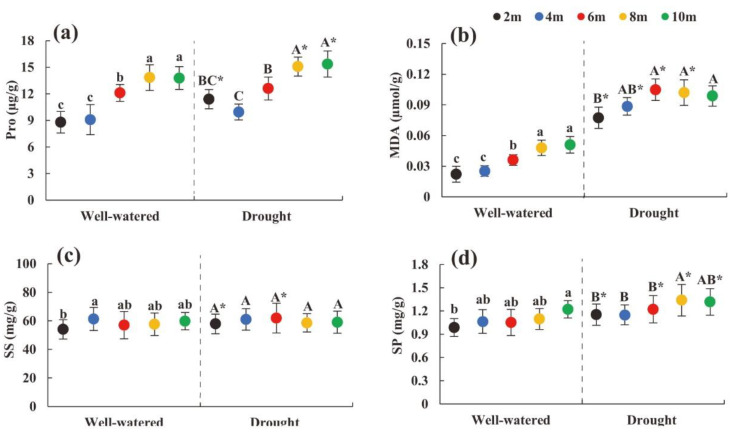
Changes in physiological characteristics of heteromorphic leaves with tree height under different soil moisture conditions. Note: (**a**–**d**) were changes in proline, malondialdehyde, soluble sugar, soluble protein content with different tree heights under different soil moisture condition. The black dot indicates the height of the 2 m tree, the blue dot indicates the height of the 4 m tree, the red dot indicates the height of the 6 m tree, the yellow dot indicates the height of the tree 8 m, and the green dot indicates the height of the 10 m tree; lowercase and uppercase letters represent the significance of the difference between different sampling heights under suitable soil moisture conditions and soil drought stress conditions, and * represents the comparison of the same height under different habitats (*p* < 0.05). Pro: proline; MDA: malondialdehyde; SS: soluble sugar; SP: soluble protein.

**Figure 5 plants-11-02376-f005:**
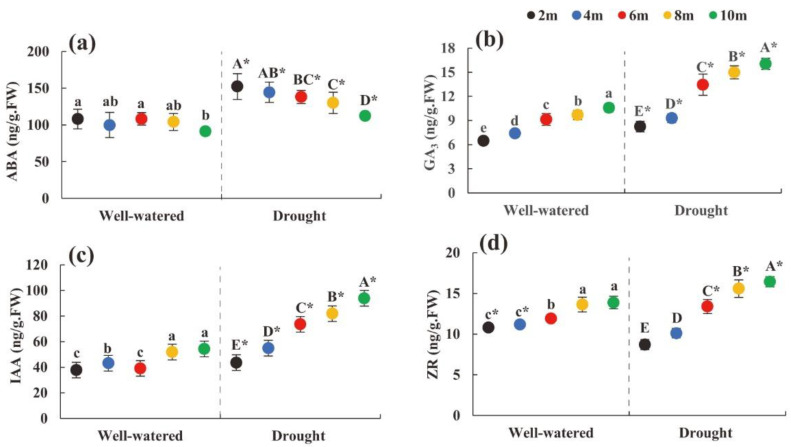
Changes in hormone content index parameters of heteromorphic leaves with tree height under different soil moisture conditions. Note: (**a**–**d**) were changes in abscisic acid, gibberellin, indoleacetic acid, zeatin riboside content with different tree heights under different soil moisture condition. The black dot indicates the height of the 2 m tree, the blue dot indicates the height of the 4 m tree, the red dot indicates the height of the 6 m tree, the yellow dot indicates the height of the tree 8 m, and the green dot indicates the height of the 10 m tree; lowercase and uppercase letters represent the significance of the difference between different sampling heights under suitable soil moisture conditions and soil drought stress conditions, and * represents the comparison of the same height under different habitats (*p* < 0.05). ABA: abscisic acid; GA_3_: gibberellin; IAA: indoleacetic acid; ZR: zeatin riboside.

**Table 1 plants-11-02376-t001:** Change in anatomical characteristics of the main veins of heteromorphic leaves with tree height under different soil moisture conditions.

Index	H (m)	Well-Watered	Drought
PT (μm^2^)	2	180.83 ± 11.24 c	211.13 ± 28.78 C
	4	200.73 ± 9.94 c	218.28 ± 38.95 C
	6	246.98 ± 39.31 b	275.18 ± 42.02 B
	8	262.67 ± 37.21 ab	284.48 ± 38.71 B
	10	293.74 ± 40.83 a	319.95 ± 40.81 A
PSR	2	2.19 ± 0.33 b	2.51 ± 0.37 C
	4	2.26 ± 0.88 b	2.53 ± 0.59 C
	6	2.37 ± 1.32 b	3.43 ± 0.78 B *
	8	3.21 ± 0.73 ab	4.31 ± 0.87 A *
	10	3.58 ± 1.05 a	4.43 ± 0.74 A *
MVBA (μm^2^)	2	42,113.66 ± 7763.62 ab	52,733.87 ± 7639.02 C *
	4	39,269.91 ± 8277.17 b	63,475.63 ± 9654.64 BC *
	6	36,605.84 ± 5670.33 b	83,282.04 ± 11,120.24 A *
	8	45,474.04 ± 9991.09 ab	85,354.90 ± 15,146.28 A *
	10	52,538.77 ± 8656.45 a	75,208.84 ± 11,081.56 AB *
MXA (μm^2^)	2	15,051.76 ± 4956.33 b	26,680.16 ± 4938.51 C *
	4	15,468.29 ± 3365.96 ab	35,882.37 ± 3677.35 B *
	6	18,103.41 ± 4764.33 ab	37,097.44 ± 2457.11 B *
	8	16,300.34 ± 2285.75 ab	49,825.14 ± 3617.41 A *
	10	19,018.17 ± 3281.91 a	45,634.07 ± 4075.40 A *
XA/VBA	2	0.34 ± 0.043 c	0.41 ± 0.042 b *
	4	0.39 ± 0.022 bc	0.41 ± 0.033 b
	6	0.34 ± 0.035 c	0.42 ± 0.025 b *
	8	0.47 ± 0.061 a	0.45 ± 0.041 a
	10	0.42 ± 0.08 ab	0.46 ± 0.027 a *
VA (μm^2^)	2	214.18 ± 33.53 b	248.74 ± 42.11 C *
	4	241.75 ± 40.19 ab	293.50 ± 59.70 C *
	6	243.28 ± 26.57 ab	311.94 ± 64.16 BC *
	8	327.87 ± 43.67 a	370.28 ± 47.12 AB *
	10	334.00 ± 38.05 a	385.34 ± 66.07 A *

Note: lowercase and uppercase letters represent the significance of the difference between different sampling heights under suitable soil moisture conditions and soil drought stress conditions, and * represents the comparison of the same height under different habitats (*p* < 0.05). PT: Palisade tissue thickness; PSR: Ratio of palisade tissue to spongy tissue; MVBA: Main vein vascular bundle area; MXA: Main vein xylem area; XA/VBA: Main vein xylem/main vascular bundle area; VA: Vessel area.

**Table 2 plants-11-02376-t002:** Regression analysis of heteromorphic leaf morphology and anatomical structure and tree height under different soil moisture conditions.

Condition	Parameter	LI	LA	LT	SLA	PT	PSR	MVBA	MXA	XA/VBA	VA
Well-watered	a	5.71	2.59	0.26	123.93	147.27	1.96	35,290.41	13,894.24	0.33	170.35
	b	−0.56	1.72	0.02	−4.07	14.89	0.14	1312.29	536.12	0.01	17.21
	R^2^	0.67	0.84	0.48	0.48	0.70	0.17	0.46	0.64	0.35	0.73
Drought	a	5.633	7.01	0.31	72.11	163.83	1.53	51,833.84	23,412.77	0.39	216.25
	b	−0.54	1.36	0.01	2.46	17.54	0.34	3363.54	2600.67	0.01	17.49
	R^2^	0.533	0.45	0.42	0.12	0.76	0.67	0.59	0.83	0.50	0.90

Note: N = 660, y = a + bx, y: tree height; a: constant; b: regression coefficient. LI: leaf index; LA: leaf area; LT: leaf thickness; SLA: Specific Leaf Area; PT: Palisade tissue thickness; PSR: Ratio of palisade tissue to spongy tissue; MVBA: Main vein vascular bundle area; MXA: Main vein xylem area; XA/VBA: Main vein xylem/main vascular bundle area; VA: Vessel area.

**Table 3 plants-11-02376-t003:** Regression analysis of the photosynthetic water physiology, physiological, and biochemical characteristics of heteromorphic leaves and tree height under different soil moisture conditions.

Condition	Parameter	Pn	Tr	Ci	RWC	LWP	Gs	WUE_i_	δ^13^C	Pro	MDA	SS	SP	ABA	GA_3_	IAA	ZR
Well- watered	a	14.94	8.10	282.79	88.80	−1.22	0.15	2.46	−28.73	7.784	0.02	51.43	0.95	116.79	5.58	32.77	9.79
	b	0.93	0.47	−4.11	−0.72	−0.05	0.01	0.01	0.07	0.552	0.01	0.83	0.02	−2.13	0.51	1.98	0.39
	R^2^	0.46	0.17	0.52	0.27	0.12	0.556	0.01	0.34	0.484	0.87	0.07	0.17	0.36	0.84	0.74	0.82
Drought	a	14.17	8.11	262.96	68.76	−2.33	0.17	1.95	−18.94	9.061	0.07	59.86	1.06	163.98	5.76	29.57	6.65
	b	1.38	0.70	−5.58	−0.05	−0.05	0.01	0.01	0.08	0.658	0.01	0.01	0.03	−4.71	1.13	6.35	1.01
	R^2^	0.53	0.43	0.50	0.01	0.16	0.21	0.01	0.53	0.507	0.14	0.01	0.17	0.75	0.66	0.92	0.64

Note: N = 660, y = a + bx, y: tree height; a: constant; b: regression coefficient; Pn: Photosynthetic rate; Tr: Transpiration rate; Gs: Stomatal conductance; C_i_: Intercellular CO_2_ concentration; RWC: Relative water content; LWP: Leaf water potential; WUE_i_: Instantaneous water use efficiency; δ^13^C: Stable carbon isotope values; Pro: Proline; MDA: Malondialdehyde; SS: Soluble sugar; SP: Soluble protein; ABA: Abscisic acid; GA_3_: Gibberellin; IAA: Indoleacetic acid; ZR: Zeatin Riboside.

## Data Availability

The data presented in this study are available on request from the corresponding author.

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
