# Peer review of "Structural and Functional Responses of the Heteromorphic Leaves of Different Tree Heights on Populus euphratica Oliv. to Different Soil Moisture Conditions"

_plants, 2022, doi:10.3390/plants11182376_

Round 1

Reviewer 1 Report (Previous Reviewer 1)

The revised version is very much improved, and is acceptable, in my opinion.

Reviewer 2 Report (Previous Reviewer 2)

I commend the authors for addressing my concerns. I have no further comments on the manuscript.

Reviewer 3 Report (Previous Reviewer 3)

All revisions have been added in the revised manuscript.

This manuscript is a resubmission of an earlier submission. The following is a list of the peer review reports and author responses from that submission.

Round 1

Reviewer 1 Report

I saw no basis for describing these as "wet" and "dry" treatments, as there was only depth to water given, and no indications of leaf water potential or water content at any height.  The leaf gas exchange data are useless, since the environmental conditions are not given.  The gas exchange data also do not make sense: in Fig. 2, lowest leaves had higher gs, equal Pn, yet lower Ci, which is not possible under uniform measurement conditions. 

The results are presented using only abbreviations for variables names.  What is "gate to sea" ratio, in leaf anatomy?  

The results seem to contradict their hypothesis rather than support it, with "dry" leaves having increases in many variables, which normally decrease in drought. 

Reviewer 2 Report

This study is important in understanding a species's structural and functional responses to water deficit conditions as affected by tree heights. The authors had exerted tremendous effort in this work, and that is admirable. Overall, the manuscript is well-written, and the story is straightforward. The references were mostly recent, and the figures and tables were clearly presented.

I have only a few comments and suggestions below to improve the manuscript.  

Line/s

Comment/s

Title

Populus euphratica’ must have complete nomenclature with the author’s initials/information

15

P. euphratica’ must be written entirely in the first mention in the abstract and in the introduction. However, on the succeeding mentions of the species, a shorter name ‘P. euphratica’ or ‘P. chinensis’ can be used.

Introduction

Add in the last paragraph the significance and uniqueness of this work. Explain how this work can contribute to this field of science.

Figure 1

It would be nice if figure legend can be inserted inside one of the graphs so readers would not go back and forth to the figure caption just to be reminded of the color identification for height differences. Otherwise, the authors may add the height values on the x-axis so it will be easy to distinguish height differences. Also, authors may opt to put a very thin broken, or straight line as a boundary between the well-watered and drought so that a division between the two treatments can be seen.

Figure 2 and other similar figures

Same comment as in Figure 1

284-294

I suggest the authors go deep into how height differences became a factor in photosynthetic, transpiration, and thus water use efficiency in heteromorphic leaves. Why do they increase as the height increases? What is driving this mechanism?

317-325

Any references to prove such claims?

Methods Section 4.2

What is the size of the plot? How many trees were being sampled?

420

‘(Zhai et al., 2020)’. Isn’t this in numerical format, just like any other references?

Conclusion

Reiterate your novelty briefly here, what this study can contribute to the growing demand for more studies relative to water use efficiency, water conservation efforts, ecosystem management, and the like. What is the take-home message? The conclusion must be as brief and concise as possible

Reviewer 3 Report

T The Authors analyzed several physiological parameteres of heteromorphic leaves of P. euphratica in two soil moisture conditions and at different heights. 

Generally, the paper contains some interesting data, and, thus, yields new information in this research area.

However, the paper should be revised as recommended, and some issues should be changed.

L61: Check “changes”

L98-98; The Authors should check the results reported in the figure 1

L104: add “different” after with

“LT” abbreviation is never explained.

In table 1 the abbreviations are reported, add also in the figures 

L120 “leave” instead of “leaf”.

L169-171 This sentence should be improved

L255, “SLA” is included in the parameters that increase as height increases while in the results it decreases.

L392 “main vein xylem area” is written two times and there are two different abbreviations near: “MXA” e “VA”.

L416: Add unit of measurement in each section of M&M

L425: Aggiungere quali saggi sono stati effettuati per valutare gli ormoni

L420 the reference “(Zhai et al., 2020)” doesn’t have the respective number in the reference

References:

L480-L498-L501L527: Check the scientific name

Check the style

Round 2

Reviewer 1 Report

The chief weakness of this paper remains the undocumented supposed "drought" treatment, which seems to be based on the depth of the water table, without any physiological measurement.
